# Short-Term Memory for Auditory Temporal Patterns and Meaningless Sentences Predicts Learning of Foreign Word Forms

**DOI:** 10.3390/brainsci12050549

**Published:** 2022-04-26

**Authors:** Elisabet Service, Erin DeBorba, Angie Lopez-Cormier, Meliha Horzum, Daniel Pape

**Affiliations:** ARiEAL Research Centre, Department of Linguistics and Languages, McMaster University, Hamilton, ON L8S 4M2, Canada; deborbaerin@gmail.com (E.D.); lopezrimd@gmail.com (A.L.-C.); meliha.horzum@gmail.com (M.H.); paped@mcmaster.ca (D.P.)

**Keywords:** phonological loop, rhythm, memory, language acquisition, second language, context signal

## Abstract

The ability to accurately repeat meaningless nonwords or lists of spoken digits in correct order have been associated with vocabulary acquisition in both first and second language. Individual differences in these tasks are thought to depend on the phonological loop component of working memory. However, phonological working memory may itself depend on more elementary processes. We asked whether auditory non-verbal short-term memory (STM) for patterns in time supports immediate recall of speech-based sequences. Participants tapped temporal sequences consisting of short and long beeps and repeated nonsense sentences sounding like their native language or an unfamiliar language. As a language learning task, they also memorized familiar-word–foreign-word pairs. Word learning was directly predicted by nonsense sentence repetition accuracy. It was also predicted by temporal pattern STM. However, this association was mediated by performance on the repetition measure. We propose that STM for temporal patterns may reflect a component skill that provides the context signal necessary to encode order in phonological STM. It would be needed to support representation of the prosodic profile of language material, which allows syllables in words and words in sentences to be ordered and temporally grouped for short-term representation and long-term learning.

## 1. Introduction

In the theoretical framework of Baddeley and Hitch [1,2], working memory (WM) is thought to consist of an attentional component (the central executive) as well as three components with storage space (the visuo-spatial sketchpad for visual and spatial information; the phonological loop for speech-based information, and the modality-general episodic buffer for event information). Of these components, the phonological loop is thought to bind and maintain verbal information for short-term representation and storage. The phonological loop is theorized to have an active component, the articulatory rehearsal process based on inner speech, and a passive phonological store component representing elements bound together in a phonological (speech-sound based) code. In other WM theories, concepts similar to the phonological loop are often referred to as verbal or phonological short-term memory (STM), making a distinction between pure storage functions and the more general WM that also encompasses processing. 

Early work with the consequences of brain lesions [3], first-language (L1) development [4], and second-language (L2) learning [5], led to the bold suggestion that the phonological loop is a language learning device [6]. Numerous studies have since shown a correlative relationship between measures of phonological loop function and both first (see e.g., [7]) and second [8,9] language acquisition. Whereas the causal relationships between language outcome measures and measures of phonological STM (pSTM) have been difficult to disentangle in L1 research [10], longitudinal studies in L2 learning have suggested that individual differences in pSTM abilities predating significant exposure to L2 predict acquisition of L2 vocabulary [11] and grammar [12]. 

In the present study, we used a meaningless-sentence repetition task to estimate pSTM capacity in young adults. The correlation of this measure to cued recall in a paired associate familiar-word–foreign-word learning task was studied in order to replicate the many previous findings of a relationship between pSTM tasks, such as non-word repetition or nonword span, and word learning. However, the main research question concerned a different task: immediate recall of a sequence of short and long auditory beeps, designed to tap STM for auditory temporal patterns. We asked the question whether STM for non-verbal temporal structure plays a role in pSTM and further in the beginning stages of learning word forms in an unfamiliar language.

Recently, a number of findings have come together to suggest that one critical aspect of pSTM functioning has to do with the ability to represent temporal order. In particular, STM for order has been associated with L1 vocabulary development in children [13], language development in monolingual and bilingual children with developmental language disorder [14,15] and dyslexia [16]. There are two prominent approaches to model the representation of order in STM in different modalities, for a review, see [17]. One assumes a gradient of item encoding strength that decreases over list positions (and possibly increases when approaching the list end) [18,19,20]. The other assumes a temporally based context signal that changes over list positions [21,22]. Items are thought to be recalled in order of activation strength, determined by either item strength or the strength of the current position-based context signal as recall unfolds. 

The effects of temporal grouping on STM have been investigated in experiments in which stimuli have been presented consecutively in groups separated by longer inter-stimulus intervals. The introduction of such temporal groups during encoding has been found to improve immediate recall of both verbal and non-verbal lists [23,24,25]. Furthermore, list position appears to be somewhat differently encoded in verbal lists with temporal grouping compared to lists in other modalities [26]. In verbal lists, group position is encoded in relation to the whole list, and item position is encoded in relation to the temporal group that the item belongs to. In other modalities, item position seems to be encoded linearly in relation to the whole list rather than the temporal group. One possibility for the representation of grouping effects for verbal lists is that a temporal structure is created for the list based on timings in stimulus presentation. Alternatively, it could be based on the activation of an internal template. The possibility that a timing pattern guides maintenance and recall receives support from a recent STM experiment [27] with instructions for participants to mentally rehearse a list of digits. Participants’ silent rehearsal was probed at different time-points, and they were asked which item they were rehearsing at that time. Participants whose rehearsal time structure more closely mimicked that of the stimulus presentation, recalled more items. A model for how a time structure in memory for auditorily presented verbal items could be spontaneously created from the list input has been proposed by Hartley, Hurlstone, and Hitch [28].

The role of rhythm and temporal duration in cognition has recently attracted growing attention in cognitive neuroscience. As data are accumulating, there has been a proliferation of hypotheses about the temporal processing machinery of the brain. Although there is wide consensus that cerebellar structures and prefrontal-striatal-hippocampal networks play a role, different proposals suggest different divisions of labor among these. In particular, the relationship between processing and remembering of absolute durations, relative order, and the relation to a regular isochronous beat has given rise to different hypothetical models [29,30,31,32]. Although the details of these models are under debate, they provide a starting point for understanding the neural implementation of a time-based context signal, as proposed in the most recent formulations of the phonological loop [33]. This can also be a new starting point for trying to understand what gives rise to those individual differences in pSTM that affect language acquisition. In the present study, we developed a new task designed to be sensitive to individual differences in the capacity of STM to present auditory temporal patterns. Performance in this task was investigated for its ability to predict pSTM and long-term learning of words in an unfamiliar language.

Laasonen and colleagues [34] studied STM for temporal patterns consisting of ordered binary sequences. Three modalities and all their combinations were studied separately. The visual stimuli were simple flashes of light (two vertically separated LEDs), the auditory stimuli were two tones with different pitches, and the tactile stimuli were touches on two different fingertips. We found the mean span length for 84% accurate comparison of pairs of sequences to be substantially correlated (*r*(44) = 0.74) with a compound measure based on five pSTM tasks. That memory for non-linguistic rhythms is related to common short-term memory tasks, such as digit span, has also been reported by other authors [35,36]. Based on these results, we constructed a comparison task of pairs of auditory tone sequences. Each sequence consisted of short and long computer-generated beeps with a constant pitch, resembling Morse code. Participants had to tap the pattern of short and long beeps from memory. This tapping measure was studied correlatively with a measure of pSTM (repetition of meaningless sentences) and a laboratory measure of word form learning (training of English-word–unfamiliar-language-word pairs for cued recall of the unfamiliar word forms when presented with the English words). 

We expected to replicate findings from the literature that pSTM predicts word learning performance. We further hypothesized that STM for temporal patterns would predict pSTM, as it would reflect the functioning of mechanisms active in providing a temporal context signal for STM for spoken material. Given these two predictions, we should also see a correlation between temporal pattern tapping and word learning as mediated by pSTM. We did not have a theory-based prediction about a direct link between STM for temporal patterns and word form learning. The relationship between the two possible predictors (temporal pattern tapping and pSTM) and word learning was investigated in a mediation analysis including the three measured variables.

## 2. Materials and Methods

### 2.1. Participants

Seventy-two participants (aged 18–48 years), recruited from McMaster University participant pools and through posters and online advertisements, participated in the study. The data of one participant were excluded because of floor level performance in the learning task. The participant sample consisted of dominant English speakers with normal or corrected-to-normal vision and self-reported normal hearing. None of the participants knew Turkish. All participants provided informed consent before beginning the experiment and were compensated for their participation by course credit or cash.

### 2.2. Tasks and Materials

#### 2.2.1. Tapping from Memory

A task to gauge short-term memory (STM) for non-linguistic temporal patterns was devised. Pure sine tones at a constant pitch (f = 527 Hz) edited in Audacity 2.3.0 (The Audacity Team–Muse Group, Limassol, Cyprus) software were used to create 10 random sequences consisting of 7 beeps, short (200 ms) and long (800 ms). The inter-stimulus interval was 200 ms. SuperLab 5.0 (Cedrus Corporation, San Pedro, CA, USA) software was used to present the sequences on an iMac computer through Sony MDRZX110NC headphones with noise-cancelling turned on. The text LISTEN was shown for 5152 ms before the auditory sequence began. After the last beep, the text REPEAT was displayed on the screen. The participant’s task was to use their index finger to tap the pattern of short and long stimuli on the computer keyboard spacebar using shorter and longer keypresses. The following trial began 8000 ms after the REPEAT prompt. The durations of the keypresses were recorded by the experimental software. 

We initially examined the mean distributions of keypress times for short vs. long stimuli in the whole group of participants. The two distributions crossed at 375 ms, at which point participants were equally likely to be responding to a short or a long beep. Based on this, taps ≤ 375 ms were scored as correct for short beeps and >375 were scored as correct for long stimuli. A second scoring was done separately for each individual participant based on their personal response time crossing points. These two methods of scoring produced nearly identical results that correlated *r*(69) = 0.97. Here we report the scores based on the individual distributions.

#### 2.2.2. Nonsense Sentence Repetition

Repetition of meaningless sentences was used as a pSTM task. Two sets of 10 sentences were presented. The first set was constructed from English 5–6-word sentences in which the content words, such as nouns and verbs, had been replaced by pseudowords adhering to English phonology and phonotactics. These nonsense sentences (“jabberwocky sentences”) sounded like English sentences. Function words, such as articles or prepositions, remained real English words. The number of syllables in each sentence varied between 6 and 8, totaling 73 syllables. The second set consisted of real Turkish sentences with 3–4 Turkish words that had either 6 or 7 syllables. The total number of Turkish syllables was 66. Speech sounds that are known to be difficult for English speakers to pronounce were avoided. All the sentence stimuli are listed in Appendix A, Table A1.

The English nonsense sentences were recorded at a 44,100 Hz sampling rate at 16 bits by a female native speaker of Canadian English using a Sennheiser ME62 omnidirectional microphone (Sennheiser, Wedemark, Germany) connected to a Focusrite Scarlett audio interface and a computer running Soundforge Pro (Magix Software GmbH, Berlin, Germany) software. The Turkish sentences were recorded by a female native speaker of Turkish using the same equipment. All recordings were made in a sound-proof room. Additionally, very low frequencies below the speech signal spectrum were filtered out with a steep 24 dB/oct high-pass filter (Waves Linear Phase EQ, Waves Audio Ltd., Tel Aviv, Israel). The recordings were normalized for loudness. The repetition task was presented by the same iMac computer that was used for the tapping task. The participants heard the sentences over earphones and were asked to immediately repeat out loud what they had heard. Their responses were recorded by the computer. Half of the participants heard the English-based stimuli first and the other half the Turkish ones.

The accuracy of the repetitions was scored by a native speaker of each language, who used the international phonetic alphabet (IPA) of each language for the broad transcription of the responses. The number of syllables that had no phoneme errors were recorded for each repeated sentence. A second score consisted of the number of word forms correctly repeated. Phoneme errors consisted of omitted, added, reordered, or replaced speech sounds (phonemes) of the language in question. A subsample of responses from 10 participants were scored by a second English speaker and a second Turkish speaker, respectively. The inter-rater reliability at the syllable level was high for both English nonsense sentences (*r* = 0.986) and Turkish sentences (*r* = 0.959).

#### 2.2.3. Foreign-Word Learning

Word form learning was operationalized as a cued recall task for foreign (Turkish) words. Participants heard English-word–Turkish-word pairs. The words were real words but not actual translations of each other. The Turkish words had three syllables which made it possible to include both words with a possible English-type stress pattern (main stress on the second syllable) and a pattern not occurring in English (main stress on the third syllable). This difference is not analyzed in the present article. The English and Turkish words were recorded by female native speakers of English and Turkish, respectively, using the same equipment and procedures as for the sentence stimuli. The word pairs were presented auditorily by an iMac computer, using the same equipment as for the tapping task. There were two lists of six pairs each (see Appendix A, Table A2). The order of the lists was counterbalanced between participants. After a pause of 3328 ms after each list presentation, the English cue words would be presented one by one, and the participant had to try to retrieve and pronounce the Turkish words that had been associated with each. When all the cue words had been presented, the same pairs were played again for four repetitions of each list. The second list was presented in the same manner. A linguistically trained native speaker of Turkish scored the responses for a number of correctly reproduced syllables using the same criteria as for the sentence repetition task. The results of 10 participants were also scored by a second linguistically trained Turkish native speaker. The correlation between the ratings was *r* = 0.985.

### 2.3. Procedure

The participants were tested in the context of three different thesis projects, each of which also included other tasks. The set of tasks was blocked and embedded in the sequence of these other tasks. The current tasks were presented in the same order for all participants: (1) English nonsense sentence repetition, (2) tapping from memory, (3) Turkish sentence repetition, (4) foreign-word learning.

## 3. Results

The scores for the three measures of interest can be seen in Table 1. For the tapping from memory task, proportion of correct short or long taps is reported. For the sentence repetition tasks, we report here the accuracy at the word form level, following much of the literature for scoring pseudoword repetition tasks. The syllable-level scoring gave similar results. The sentence repetition scores for English nonsense sentences and Turkish sentences were correlated *r*(69) = 0.595 (CI: 0.42–0.727). The scores for proportion of correctly repeated words or pseudowords in English nonsense sentences and Turkish sentences were averaged. This produced a distribution that did not significantly differ from normality (Shapiro–Wilk’s test). For the learning task, word-level scoring produced floor effects. We, therefore, report proportion of correctly recalled syllables on the fourth and final trial. 

In our analysis, we first inspected correlations between all three measures. We then studied a mediation model for support of the hypotheses (1) that memory for temporal patterns predicts ability to form STM representations of meaningless verbal material and (2) that STM for such material, as measured by the present sentence repetition tasks, facilitates its learning for LTM. We also explored the possibility that there would be a direct link between STM for temporal patterns and vocabulary learning. The correlation matrix for the three measures is shown in Table 2.

All correlations between the three variables were significant. Our final analysis addressed the question of whether the link between tapping and word learning is mediated by phonological STM, here operationalized as meaningless-sentence repetition. The results of the mediation analysis run on R scripts provided in the *medmod* module by the jamovi project [37] are shown in Figure 1. This figure displays the path coefficients between the different variables in the mediation model. The path from tapping to nonsense sentence repetition has a significant coefficient. The path from nonsense sentence repetition to foreign-word learning is also significant. However, the direct path from tapping to foreign-word learning is not significant after the mediation of the nonsense sentence repetition has been considered. The analysis suggests that individual differences in STM for temporal patterns share variance with phonological STM, which in turn shares variance with long-term learning of new word forms. However, no significant direct path from STM for temporal patterns remained when the mediating effect of phonological STM had been accounted for. The estimated total variance accounted for by the model was 53.9%.

## 4. Discussion

This study set out to explore whether individual differences in STM for the temporal structure of auditory information predict the capacity of phonological STM, and whether there is an indirect connection to the learning of new word forms in an unfamiliar language. We studied three tasks: tapping temporal patterns from STM, repetition of meaningless sentences, and learning of new unfamiliar-sounding word forms coupled with familiar native-language words. These tasks were used as measures of STM for auditory patterns in time, pSTM, and aptitude for memorizing word forms in a new language, respectively. We replicated former studies in finding a significant correlation between measures of pSTM and word form learning. We also found a significant correlation between the tapping measure of STM for temporal patterns and the ability to repeat meaningless sentences. This finding was also hypothesized based on previous literature [34,35,36]. Further, the correlation between tapping and word learning was positive and significant. However, a mediation analysis suggested that this correlation was mediated by pSTM. These findings are significant from the point of view of understanding how the phonological loop may function as a language learning device [6]. They also suggest that STM for temporal patterns may form a necessary part of STM for material that changes in time. The tapping task studied here reflects, perhaps, the behavior of a context signal that plays a central role in STM for temporal order.

One prominent question among the working memory researchers has been the modality specificity of STM for order [17]. The present study only explored STM for temporal order in the auditory modality. However, the finding of a robust correlation between a pSTM task and a non-verbal task gauging STM for temporal patterns is very similar to our previous findings of short-term recognition memory for temporal structure being related to five pSTM tasks [34]. In the previous work, three modalities and their combinations were tested. What is common to those previous tasks and the present task is the binary nature of the temporal sequences. With only two kinds of stimuli in each sequence, memory for individual items can contribute very little to memory for the whole sequence. Other studies that have reported a correlation between STM for rhythm and verbal STM have typically varied only the length of unfilled intervals between stimuli [35,36]. Thus, in these types of tasks, all the information is embedded in the temporal sequence of binary stimuli, be they of different duration (e.g., short and long), different perceptual quality (e.g., high and low tones) or preceded by a shorter or longer pause. What appears to be shared by the tasks is a representation of information unfolding in time that cannot be readily represented as a path in space. Temporal order, possibly based on both absolute duration and relative position, handled by interacting brain networks [30,31], may be an essential component of representations created by the phonological loop. The temporal dimension would be bound to the phonological information related to language networks and motor networks that support articulation.

From the point of view of language acquisition, the present study suggests that aspects of the temporal structure of the language stimulus are important for its retention. Not only the prosodic patterns of stress, pitch, and length of syllables, but also the sequencing of phonemes within syllables, syllables within words, words within phrases, etc., may be an important part of temporally structured language acquisition. Both typical and atypical language may benefit from interventions that support the detection, representation, and rehearsal of rhythmic patterns in language [38].

## Figures and Tables

**Figure 1 brainsci-12-00549-f001:**
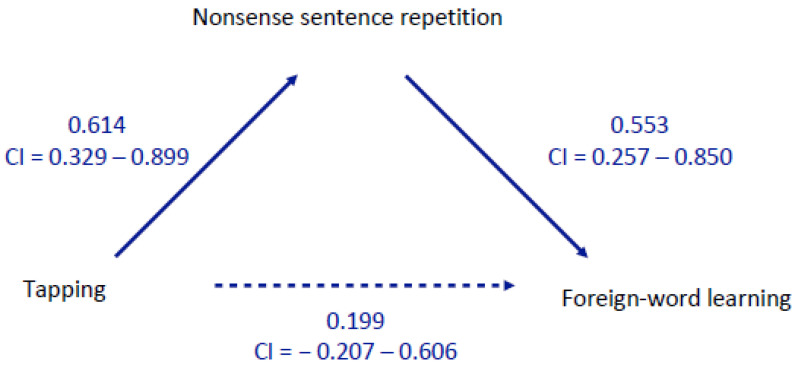
Mediation model showing the direct path from tapping to foreign-word learning and the indirect path from tapping to foreign-word learning as mediated by nonsense sentence repetition. CI = 95% confidence interval. Paths for which the confidence interval includes zero are not considered reliable.

**Table 1 brainsci-12-00549-t001:** Mean scores for tapping from memory (proportion of correct tap durations), for nonsense sentence repetition (proportion of words/pseudowords correct), for foreign-word recall on the fourth trial (proportion of syllables correct), and Shapiro–Wilk’s test of normality of the distributions.

	Mean Score (SD)	Range	Shapiro- Wilk’s	Shapiro-Wilk’s *p*
**Tapping**	0.764 (0.103)	0.443–0.971	0.981	0.361
**Nonsense sentence repetition**	0.545 (0.142)	0.164–0.875	0.985	0.54
**Foreign-word learning**	0.386 (0.185)	0.056–0.889	0.973	0.128

**Table 2 brainsci-12-00549-t002:** Pearson’s correlations between tapping (N = 71), nonsense sentence repetition (N = 71) and foreign-word learning (N = 71) scores and 95% confidence intervals (CI).

	Tapping	Nonsense Sentence Repetition	Foreign-Word Learning
**Tapping**	1	0.448 *** (CI = 0.240–0.614)	0.302 * (CI = 0.074–0.500)
**Nonsense sentence repetition**		1	0.474 *** (CI = 0.271–0.637)
**Foreign-word learning**			1

* *p* < 0.05, *** *p* < 0.001.

## Data Availability

The data presented in this study are openly available in the McMaster Dataverse repository. Rep-lication data for Service_Brain_Sci_2022_12_549 (view at https://dataverse.scholarsportal.info/dataset.xhtml?persistentId=doi:10.5683/SP3/XLRXBA) was published in McMaster University Dataverse (view at https://dataverse.scholarsportal.info/dataverse/mcmaster).

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
