# Peer review of "Short-Term Memory for Auditory Temporal Patterns and Meaningless Sentences Predicts Learning of Foreign Word Forms"

_brainsci, 2022, doi:10.3390/brainsci12050549_

Round 1

Reviewer 1 Report

The paper aims to examine whether nonverbal auditory short-term memory (STM) for temporal structure supports immediate recall of speech-based sentences through phonological STM and subsequently word learning in an unfamiliar language. Results showed that word learning was indeed predicted by both nonsense sentence repetition accuracy and temporal pattern in short-term memory, and that the latter relationship was mediated by performance on the repetition measure. These findings shed new light on the functioning of phonological loop and highlight the important role of temporal order as a constitutive component of representations created by the phonological loop.

The manuscript is clear, very well-written and well structured. I just have a difficulty of understanding in the Introduction p 2. I’m not sure I understand what the authors mean by “temporal groups”, could they tell us more.  

The method used appears to be appropriate for testing the hypotheses. The different tasks are well explained, and the authors have also provided the stimuli for the nonsense sentence repetition and foreign-word learning tasks. However, not being familiar with the tapping task, I am not sure how participants can type the sequence of sound stimuli on the computer keyboard - I mean do they have to turn the sound into a verbal musical note? In addition, for the sake of reproducibility, I would recommend adding a procedure section explaining the context and organization of the testing session (task order, counterbalancing between tasks, breaks between the tasks…).

The figure and the tables are very good and easy to interpret. The statistical analyses seem sound.

Finally, conclusions are very consistent with the evidence and arguments presented.

I encourage the authors to make their data freely available to promote open science practices.

Author Response

Thank you for your supportive comments. Replies are below.

Comment 1. 

The manuscript is clear, very well-written and well structured. I just have a difficulty of understanding in the Introduction p 2. I’m not sure I understand what the authors mean by “temporal groups”, could they tell us more.

Reply 1: Clarifying text has been added to page 2.

Comment 2: 

The method used appears to be appropriate for testing the hypotheses. The different tasks are well explained, and the authors have also provided the stimuli for the nonsense sentence repetition and foreign-word learning tasks. However, not being familiar with the tapping task, I am not sure how participants can type the sequence of sound stimuli on the computer keyboard - I mean do they have to turn the sound into a verbal musical note? In addition, for the sake of reproducibility, I would recommend adding a procedure section explaining the context and organization of the testing session (task order, counterbalancing between tasks, breaks between the tasks…).

Reply 2: The task description has been expanded in the last paragraph of page 3. A Procedure section has been added to page 5.

Comment 3: 

I encourage the authors to make their data freely available to promote open science practices.

Reply 3: We have added information on the repository where the data file will be available if the article is accepted to page 7.

Reviewer 2 Report

Service and colleagues reported a study in which they used an individual difference approach to examine relations among the auditory non-verbal short-term memory capacity for temporal order, phonological short-term memory capacity, and unfamiliar language learning ability. To measure auditory non-verbal STM capacity for temporal order, they used an auditory STM tapping task. To measure phonological STM, they used a nonsense sentence repetition task. To measure the ability of unfamiliar language learning, they used a foreign word learning task. Specifically, they tested whether individual differences in order STM predict the phonological STM, and whether there is an indirect relation to the new word learning in an unfamiliar language.

They conducted correlational analyses among these three measures, and found that there were significant correlations among all pairs of variables, though the significance for the relation between order STM and language learning is much less than the other two.  In addition, authors conducted a mediation analysis and found that relation between order STM and language learning disappeared when the phonological STM was accounted for. Based on these findings. the researchers claimed that the ability of retention for temporal structure is an essential component of representations create by the phonological loop, which binds contextual information with verbal materials, hence benefits new language learning.

This study examined an important topic, and provided a valuable case exploring the underlying component which links the phonological STM and language learning. The conceptual framing and analyses are straightforward, and the results are sounding. I believe this work made a good contribution to the field. I only have a few concerns that I hope the authors could address.

  1. It was well known that phonological STM predicted word learning by previous work, and this was well replicated by the current study. If the authors want to explore the underlying mechanism, and test whether the temporal order STM creates a contextual representation for verbal materials, and whether such capacity associates with language learning, it feels like that the order STM should be regarded as a mediator in the mediation analysis, instead of using phonological STM. I am not an expert of mediation analysis, so this concern may reflect my poor understanding of mediation analysis.
  2. What software or package was used for the mediation analysis, and what kind of statistical testing was used?
  3. It is known that the correlational analyses are sensitive to outliers. According to the ranges of three variables in the description results, there seemed large variances across subjects. Are there any outliers in the data? I think including scatterplots would make the description results more clear.

Author Response

Replies to comments are below.

Comment 1:

It was well known that phonological STM predicted word learning by previous work, and this was well replicated by the current study. If the authors want to explore the underlying mechanism, and test whether the temporal order STM creates a contextual representation for verbal materials, and whether such capacity associates with language learning, it feels like that the order STM should be regarded as a mediator in the mediation analysis, instead of using phonological STM. I am not an expert of mediation analysis, so this concern may reflect my poor understanding of mediation analysis.

Reply 1: Our main research question of interest was whether individual differences in STM for the temporal structure of non-verbal material is related to individual differences in phonological STM. As phonological STM has been suggested to play a role in vocabulary acquisition, we were also curious about the possibility that there would be a correlation between STM for temporal structure and learning foreign words. We chose repetition of meaningless sentences for our pSTM repetition task to include as many aspects of the temporal structure of real language as possible. However, as previous studies have used single pseudowords or lists of digits, words or pseudowords, we needed to test that our sentence repetition task is also correlated to word learning. In the first step, we explored correlations. We found robust correlations between nonsense sentence repetition and the tapping from memory task. There was also a correlation with a smaller effect size between tapping and word learning. The mediation model tested whether a direct path from tapping to word learning was reliable if mediation by nonsense sentence repetition was taken into account. This connection was no longer reliable in our sample (but still might be in a larger sample). So, we did not really ask the question whether the relation between phonological STM and word learning might be explained by memory for temporal patterns. We do not really believe that the unique variance in memory for temporal structures is related to word learning. Instead, we do think that phonological STM serves to bind together language material such as phonemes, syllables etc. This is probably crucial for language acquisition. What we think our study means is that STM for temporally structured information may be necessary for phonological STM to do its job. However, a correlational study cannot give a definite answer, so we will need experimental evidence. We do not think that the alternative mediation model could answer our questions. We have added a little text to try to better explain our starting point.

Comment 2: 

What software or package was used for the mediation analysis, and what kind of statistical testing was used?

Reply 2: We have now made it more visible in the text that we used the medmod module of the openly available jamovi project. A link to it is listed in the list of references. Basically, these models are regressions.

Comment 3:

It is known that the correlational analyses are sensitive to outliers. According to the ranges of three variables in the description results, there seemed large variances across subjects. Are there any outliers in the data? I think including scatterplots would make the description results more clear.

Reply 3: Our dataset had hardly any real outliers. Excluding all participants that were outside two standard deviations of the mean of any of the three variables only resulted in two or three individuals being excluded depending on the correlation. Of these, most exclusions were based on small differences in the third decimal, so these data points were not real outliers. The analysis with these exclusions did not change any of the statistical findings. We can add scatterplots if the editor thinks it is helpful (sorry I cannot upload to this online response).